



**Diagnosis toward predicting mean annual runoff in ungauged basins**
Yuan Gao, Lili Yao, Ni-Bin Chang and Dingbao Wang[*]
Department of Civil, Environmental, and Construction Engineering, University of Central
Florida, Orlando, FL 32816, United States
[*]Correspondence to D. Wang, dingbao.wang@ucf.edu
**Abstract**
Prediction of mean annual runoff is of great interest but still poses a challenge in ungauged basins.
The present work diagnoses the prediction in mean annual runoff affected by the uncertainty in
estimated distribution of soil water storage capacity. Based on a distribution function, a water
balance model for estimating mean annual runoff is developed, in which the effects of climate
variability and the distribution of soil water storage capacity are explicitly represented. As such,
the two parameters in the model have explicit physical meanings, and relationships between the
parameters and controlling factors on mean annual runoff are established. The estimated
parameters from the existing data of watershed characteristics are applied to 35 watersheds. The
results showed that the model could capture 88.2% of the actual runoff on average, indicating that
the proposed new water balance model is promising for estimating mean annual runoff in
ungauged watersheds. The underestimation of runoff is mainly caused by the underestimation of
the spatial heterogeneity of soil storage capacity due to neglecting the effect of land surface and
bedrock topography. A higher spatial variability of soil storage capacity estimated through the
Height Above the Nearest Drainage (HAND) indicated that topography plays a crucial role in
determining the actual soil water storage capacity. The performance of mean annual runoff
prediction in ungauged basins can be improved by employing better estimation of soil water
storage capacity including the effects of soil, topography and bedrock. The purpose of this study



is to diagnose the data requirement for predicting mean annual runoff in ungauged basins based
on a newly developed process-based model.
**Keywords**: mean annual runoff; ungauged; storage capacity; curve number; soil; topography;
bedrock
**1.    Introduction**

Hydrologists have a long-standing interest in mean annual water balance modeling and

prediction.  The factors controlling mean annual runoff have been studied in the literature.  Mean
climate has been identified as the first order control on mean annual runoff and evaporation and it
has been quantified by climate aridity index, which is defined as the ratio between the mean annual
potential evapotranspiration and precipitation (Turc, 1954; Pike, 1964).  Other controlling factors
include the temporal variability of climate (Farmer et al., 2003; Troch et al., Fu and Wang, 2019),
vegetation (Zhang et al., 2001; Donohue et al., 2007; Gentine et al., 2012; Li et al., 2013), soil
(Atkinson et al., 2002; Yokoo et al., 2008; Li et al., 2014), and topography (Woods, 2003;
Abatzoglou and Ficklin, 2017).  Mean annual runoff or evaporation has been modeled as a function
of climate aridity index and the equation is usually called as Budyko equation (Budyko, 1958).
The effects of other factors are represented by including a parameter to Budyko equations (Fu,
1981; Yang et al., 2008; Wang and Tang, 2014).  Among these factors, climate including its mean
and temporal variability, and soil water storage capacity including its mean and spatial variability
are dominant catchment characteristics controlling mean annual runoff, especially for saturation
excess runoff generation-dominated catchments (Milly, 1994).

Intra- and inter-annual climate variability introduces non-steady state conditions to finer

timescale water balances and the non-steady state effect could propagate to the mean annual runoff.



The effects of seasonal variations of precipitation and potential evaporation on long-term runoff
have been studied in several studies.  Milly (1994) showed that seasonality tends to increase mean
annual runoff through a stochastic soil moisture model.  The seasonality effects have been
demonstrated through a top-down model by Hickel and Zhang (2006) and a classification study by
Berghuijs et al. (2014).  Mean annual water balance also receives impacts from climate variability
at the inter-annual and daily timescales.  Li (2014) showed that the inter-annual variability of
precipitation and potential evaporation could increase the mean annual runoff up to 10% based on
a stochastic soil moisture model.  Shao et al. (2012) found that daily precipitation with a larger
variation potentially increases mean annual runoff especially in the catchments where infiltration
excess runoff is prevalent.  Yao et al. (2020) quantified the relative contribution of daily, monthly
and inter-annual climate variabilities to mean annual runoff and showed that the contribution
decreases, by average, from monthly to inter-annual scale, and then daily scale.

Soil water storage capacity exerts a powerful control on mean annual runoff.  A smaller

soil water storage capacity creates favorable conditions for runoff generation because the
precipitation in excess of the available storage capacity would be lost as runoff directly, while
catchments with a lager soil water storage capacity could hold more precipitation for evaporation
(Sankarasubramanian and Vogel, 2002; Porporato et al., 2004; Chen et al., 2013).  Soil water
storage capacity is closely related to vegetation since the root structure of vegetation could affect
soil water holding capacity significantly.  Research has been conducted to reveal the role of soil
water storage capacity through the linkage of vegetation and model parameter (Yang et al., 2008;
Chen and Wang, 2015). Gerrits (2009) developed equations for transpiration and interception by
considering the root zone and interception storage capacity as two of the most important catchment
characteristics affecting evapotranspiration.  In addition to the magnitude of the average soil water





storage capacity, the spatial variability of storage capacity within a catchment also influences
precipitation partitioning at the event scale, and further influences the cumulative runoff at the
mean annual scale (Moore, 1985; Jothityangkoon et al., 2001; Gao et al., 2016).  It has also been
suggested that the spatial variability of soil water storage capacity could suppress the actual
evaporation and therefore promote the runoff generation indirectly (Yao et al., 2020).

Therefore, climate variability and soil water storage capacity need to be explicitly

incorporated into the model for predicting mean annual runoff.  The effect of climate variability
could be taken into account by driving the model with daily precipitation and potential evaporation
which are usually available.  The spatial distribution of soil water storage capacity could be
modelled by a distribution function, and it is usually modelled by the generalized Pareto
distribution (Moore, 1985; Zhao, 1992).  The distribution function includes two parameters, i.e.,
the shape parameter and the maximum storage capacity over the watershed.  In ungauged basins,
soil water storage capacity and its spatial variability need to be estimated directly from available
data.  Gao et al. (2014) adopted the mass curve technique, which has been used for designing the
storage capacity of reservoir, to estimate the average water storage capacity of the root zone using
precipitation and potential evaporation data.  The shape parameter of the distribution function has
been estimated from soil data (Huang et al., 2003).  However, the estimated parameters from these
methods bring much uncertainty in runoff estimation, and the two parameters of the generalized
Pareto distribution are usually estimated by model calibration using observed streamflow data
(Wood et al., 1992; Alipour and Kibler, 2018, 2019).

The objective of this paper is toward developing nonparametric mean annual water balance

model for predicting mean annual runoff in ungauged basins, which remains a challenge for
hydrologists (Blöschl et al., 2013). The mean annual water balance model is forced by daily



precipitation and potential evaporation; therefore, the climate variability at different timescales is
represented explicitly in the climate input. The runoff generation is quantified by a distribution
function for describing the spatial distribution of soil water storage capacity (Wang, 2018). The
mean and the shape parameter of the distribution function need to be estimated from the available
data in ungauged basins. Therefore, the model serves as a diagnosis tool for evaluating the data
requirement for estimating soil water storage capacity. The mean of the distribution is estimated
from curve number and climate since the distribution function leads to the SCS curve number
method. The estimation of the shape parameter is diagnosed in terms of the data requirement
including soil, land surface topography, and bedrock topography. Section 2 introduces the new
mean annual water balance model and the study watersheds. Results and discussion are presented
in Section 3, followed by Section 4 for conclusions.
**2. Methodology**
**2.1 Mean annual runoff model**
As discussed in the introduction, the mean annual runoff model takes daily precipitation
and potential evaporation as inputs, and calculates daily soil wetting (infiltration) and evaporation
by tracking the soil water storage. Mean annual runoff is estimated by aggregating the daily values.
The daily soil wetting is calculated using the concept of saturation excess runoff generation by
modeling the spatial variability of soil moisture and storage capacity. To facilitate the parameter
estimation of storage capacity distribution in ungauged basins, the following distribution function
is used for modeling the spatial distribution of storage capacity (Wang, 2018):

$$F(C) = 1 - \frac{1}{a} + \frac{C + (1-a)S_b}{a\sqrt{(C+S_b)^2 - 2aS_bC}} \qquad (1)$$



where $F(C)$ is the cumulative distribution function (CDF), representing the fraction of the
watershed area for which the storage capacity is equal to or less than $C$; $a$ is the shape parameter
of the distribution and varies between 0 and 2; and $S_b$ is the average soil water storage capacity
over the watershed (i.e., the mean of the distribution). As shown in Wang (2018), this distribution
function leads to the SCS curve number (SCS-CN) method when the initial storage is set to zero.
Therefore, there is a linkage between $S_b$ and the "potential maximum retention after runoff begins"
in the SCS-CN method, denoted as $S_{CN}$.

Daily soil wetting and runoff generation is computed as a function of daily precipitation

($P$), initial storage ($S_0$), $a$, and $S_b$. As shown in Wang (2018), the average soil wetting ($W$) is
computed by:

$$W = \frac{P + S_b\sqrt{(m+1)^2 - 2am} - \sqrt{[P + (m+1)S_b]^2 - 2amS_b{}^2 - 2aS_bP}}{a} \qquad (2)$$

where $m = \frac{S_0(2S_b - aS_0)}{2S_b(S_b - S_0)}$. Setting $S_0 = 0$ and dividing $P$ on both sides of equation (2), a Budyko-
type equation, representing $\frac{W}{P}$ as a function of $\frac{S_b}{P}$, is obtained (Wang and Tang, 2014), which has
been used to model long-term soil wetting (Tang and Wang, 2017). Therefore, equation (2) can
be interpreted as a non-steady state Budyko equation which accounts for the effect of water storage.
Daily evaporation is computed as (Yao et al., 2020):

$$E = \frac{W + S_0}{S_b} \frac{E_p + S_b - \sqrt{(E_p + S_b)^2 - 2aS_bE_p}}{a} \qquad (3)$$

The first component on the right-hand side of equation (3), $\frac{W + S_0}{S_b}$, is the percentage of storage, and
the second component is the evaporation for the condition when the entire watershed is saturated,
i.e., the spatial distribution of soil water storage is same as that of storage capacity (Yao et al.,
2020). Dividing $W + S_0$ on both-hand sides, equation (3) represents $\frac{E}{W + S_0}$ as a function of $\frac{E_p}{S_b}$, and





the function is same as the Budyko-type equation derived by Wang and Tang (2014). Mean annual
evaporation ($\bar{E}$) is computed by aggregating the daily evaporation, and mean annual runoff ($\bar{Q}$) is
computed as the difference of mean annual precipitation and evaporation.

This mean annual water balance model applies two non-steady Budyko-type equations at

the daily scale, one for daily soil wetting and the other for daily evaporation. Runoff routing is
not necessary since the model is for long-term water balance. As a result, the mean annual water
balance model includes two parameters, i.e., the shape parameter ($a$) and the average soil water
storage capacity ($S_b$). For studies where a one-parameter Budyko equation is applied to long-term
scale directly, the effects of climate variability (seasonality, inter-annual variability, and daily
storminess) on mean annual water balance are attributed to the single parameter of Budyko
equation (e.g., Fu, 1981; Zhang et al., 2001). This creates the challenge to estimate the single
parameter in ungauged basins; whereas, the mean annual water balance model used in this paper
takes daily precipitation and potential evaporation as inputs, and the effects of climate variability
are taken into account explicitly. To achieve the goal of predicting mean annual runoff in
ungauged basins, $a$ and $S_b$ need to be estimated in ungauged basins.
**2.2 Parameter estimation**
**2.2.1   Average soil water storage capacity**

Under a given soil moisture condition, soil water storage capacity is the sum of actual water

storage and the remaining (or effective) storage capacity. The effective storage capacity
corresponding to the normal antecedent moisture condition defined in the SCS-CN method, $S_{CN}$
(mm), is computed as a function of CN (SCS, 1972; Bartlett et al., 2016):

$$S_{CN} = 25.4(1000/CN - 10) \tag{4}$$





where CN is computed based on land use and land cover (LULC) and hydrologic soil group (HSG)
for each catchment. The LULC data can be obtained from the National Land Cover Database
(Homer et al., 2015), and the HSG data can be extracted from the Gridded Soil Survey Geographic
(gSSURGO) database with a spatial resolution of 10 m (*USDA*, 2014). In HSG, soils are assigned
to one of the four groups (A, B, C, and D) and three dual classes (A/D, B/D, and C/D) according
to the rate of infiltration when the soils are not protected by vegetation and receive precipitation
from long-duration storms. For the cells characterized by dual classes, the CN value is calculated
as the average of the two CN values corresponding to the two soil groups.

The average soil water storage capacity ($S_b$) is the sum of the actual storage under the

normal condition ($\bar{S}$) and its corresponding effective storage capacity:
$$S_b = \bar{S} + S_{CN} \tag{5}$$

Since the "normal antecedent moisture" can be interpreted as the steady-state soil moisture
condition, $\bar{S}$ is the long-term average storage over the watershed. The values of $\bar{S}$ for 59 MOPEX
(MOdel Parameter Estimation Experiment) watersheds are estimated based on the long-term water
balance model in Yao et al. (2020); and these watersheds do not include any watersheds studied in
this paper. The long-term water balance model used in their study has a same model structure but
the two parameter, i.e., the mean value of the soil water storage capacity and its shape parameter
in the distribution function, were obtained by model calibration. The ratio between $\bar{S}$ and $S_b$ is
defined as the long-term storage ratio $\left(\frac{\bar{S}}{S_b}\right)$. It is found that the values of $\frac{\bar{S}}{S_b}$ for all the watersheds
were larger than 0.5. As shown in Figure 1, $\frac{\bar{S}}{S_b}$ has a linear relationship with the climate aridity
index:
$$\frac{\bar{S}}{S_b} = -0.46\Phi + 1.2 \tag{6}$$



where $\Phi$ is the climate aridity index. Substituting equations (5) and (6) into equation (4), one can
estimate the average soil water storage capacity as a function of curve number and climate aridity
index:
$$S_b = \frac{S_{CN}}{0.46\Phi - 0.2}$$   (7)
**2.2.2   Shape parameter**
The spatial variability of storage capacity is determined by the spatial distribution of point-
scale pore space across the watershed. The volume of soil pores at point scale can be determined
by soil thickness and porosity in different soil layers. The porosity ($\theta_s$) for each layer is calculated
from the soil bulk density:
$$\theta_s(j) = 1 - \frac{\rho_b(j)}{\rho}$$   (8)
where $j$ denotes the $j^{th}$ soil layer; $\rho_b(j)$ is the bulk density of the $j^{th}$ soil layer; $\rho$ is the particle
density (2.65 g/cm$^3$). After obtaining the porosity, the point-scale storage capacity can be
calculated as the following equation (Huang et al., 2003):
$$C = \sum_1^n z_j \cdot \theta_s(j)$$   (9)
where $C$ is the point-scale soil storage capacity; $n$ is the number of soil layers; $z_j$ and $\theta_s(j)$ are the
thickness and porosity of the $j^{th}$ soil layer, respectively. In the gSSURGO database, the soil
thickness and bulk density for each layer are available for shallow soil from the land surface to ~
2 m soil depth.
The total soil thickness at each point is the elevation difference from the land surface to the
fresh bedrock. However, the bedrock topography is difficult to obtain especially at the catchment
scale. Alternatively, it is assumed that the spatial distribution of the actual soil water storage
capacity is same as the spatial distribution of water storage capacity computed from the gSSURGO
database. In order to compare the shape parameter evaluated from the soil data with its





counterparts evaluated from other methods, the point-scale storage capacity is normalized with the
average storage capacity over the watershed, and Equation (1) is rewritten as:
$$F(x) = 1 - \frac{1}{a} + \frac{x + (1-a)}{a\sqrt{(x+1)^2 - 2ax}}$$    (10)
where $x$ is the normalized storage capacity $\left(\frac{C}{S_b}\right)$ at point scale; $a$ is the shape parameter describing
the spatial variability of soil water storage capacity.  The shape parameter $a$ is then estimated
through fitting the point-scale storage capacity data obtained from Equation (9) by minimizing the
root mean square error (RMSE).
**2.3. Study watersheds**

The estimations of mean annual runoff in 35 watersheds are diagnosed in this paper.  The

drainage area of the watersheds varies from 2044 to 9889 km$^2$.  Table 1 shows the USGS gauge
number and climate aridity index of these watersheds.  The human interferences are minimum
(Wang and Hejazi, 2011), and saturation excess is the dominated runoff generation in these
watersheds.  Daily precipitation and streamflow data during 1948 – 2003 are extracted from the
MOPEX dataset (Duan et al., 2006), and the daily potential evaporation during this period is
calculated based on the Hargreaves method (Hargreaves and Samani, 1985) by using the daily
maximum, minimum, and mean temperature.  The average soil water storage capacity and the
shape parameter for these watersheds are estimated from the available data of climate, LULC, soil,
and topography, and the predictions of mean annual runoff are diagnosed.
**3.   Results and discussion**
**3.1. Estimated average soil water storage capacity**

The potential maximum retention ($S_{CN}$) is calculated based on the average CN in each

watershed (Table 1).  The average CN is computed based on LULC and hydrologic soil group.




For examples, Figure 2a shows the LULC map for the Fox River watershed in Wisconsin and
Figure 2d shows the LULC map for the Spoon River watershed in Illinois. The dominant land
uses are agriculture (49%) and forest (33%) in the Fox River watershed, and agriculture (77%) and
forest (15%) in the Spoon River watershed. The hydrologic soil groups are shown in Figure 2b
(Fox River watershed) and Figure 2e (Spoon River watershed). Given the same LULC, the
hydrologic soil group D is more favorable for runoff generation compared with group A. The
dominant hydrologic soil groups are group A (31%) and group B (19%) in the Fox River watershed,
and group C/D (49%) and group B/D (20%) in the Spoon River watershed. The calculated CN for
each grid cell is shown in Figure 3c (Fox River watershed) and Figure 3f (Spoon River watershed).
The average CN is 61.0 for the Fox River watershed and 78.1 for the Spoon River watershed.
Since the Spoon River watershed has a higher percentage of agricultural land and lower soil
permeability, its average CN is higher than that for the Fox River watershed. Correspondingly,
the calculated $S_{CN}$ in the Fox River watershed (162 mm) is higher than that in Spoon River
watershed (71 mm). The values of $S_{CN}$ over the study watersheds vary from 56 mm (Auglaize
River watershed) to 182 mm (Chattahoochee River watershed) as shown in Table 1.

The average soil water storage capacity is estimated based on the computed $S_{CN}$ and

climate aridity index shown in Equation (7). For examples, the climate aridity index in the Fox
River watershed is 1.12 which is the same as that in the Spoon River watershed. The estimated $S_b$
is 721 mm in the Fox River watershed and 314 mm for the Spoon River watershed. As shown in
Table 1, the estimated $S_b$ varies from 177 mm (Chikaskia River watershed) to 1870 mm
(Chattahoochee River watershed) over the study watersheds. Figure 3a shows the spatial
distribution of the estimated $S_b$. Watersheds with higher $S_b$ are mostly distributed in the eastern
US, where the aridity index is relatively lower than that in the other watersheds.



### 3.2. Estimated shape parameter


The shape parameter ($a$) for the distribution of soil water storage capacity is estimated
based on the soil data in the gSSURGO database. For examples, the black circles in Figure 4 show
the normalized storage capacity for the Fox River watershed (Figure 4a) and the Spoon River
watershed (Figure 4b) based on the soil data in the gSSURGO database. As shown in Figure 4,
the normalize CDF for both watersheds shows an S-shape. The estimated shape parameter is 1.996
for the Fox River watershed (RMSE = 0.58) and 1.990 for the Spoon River watershed (RMSE =
1.27) by fitting to the soil data. Higher value of shape parameter indicates less spatial variability;
therefore, the spatial variability in the Spoon River watershed is higher than that in the Fox River
watershed. The mean value of RMSE for the 35 study watersheds is 0.06. Figure 3b shows the
estimated shape parameters for the study watersheds, which vary from 1.830 to 1.998.

### 3.3. Diagnosing mean annual runoff prediction


The estimated values of $S_b$ and $a$ based on climate, LULC, and soil data are applied to the
mean annual water balance model. The comparison of simulated and observed mean annual runoff
for the study watersheds is shown in Figure 5a. The RMSE for estimated mean annual runoff is
80 mm/yr. The water balance model captures 88.2% of the mean annual runoff; therefore, the
methods for estimating $S_b$ and $a$ based on the available data are promising for predicting annual
runoff in ungauged basins.
The water balance model with the estimated values of $S_b$ and $a$ underestimates the mean
annual runoff in some watersheds, and the relative underestimation error is 11.8% on average
among all the study watersheds. The underestimation of mean annual runoff could be due to the
biased estimation of the shape parameter. As described in Section 3, the spatial variability of soil
storage capacity is assumed to be equal with the spatial variability of the pore space in the shallow





soil.  The pore space at the point scale is calculated through the porosity and soil thickness.  The
thickness of the shallow soil in the gSSURGO database is quite uniformly distributed across the
watershed, i.e., around 2 m; whereas, the actual soil thickness including the weathered bedrock is
the elevation difference between the land surface and fresh bedrock, and can be highly
heterogeneous due to the variable land surface and bedrock topography over the catchment.

To diagnose the effect of land surface and bedrock topography on mean annual water

balance, the shape parameter is calibrated using the observed streamflow.  The streamflow data
during 1948-2003 are divided into three periods: 1) the warm-up period (1948-1953); 2) the
calibration period (1954-1973); and 3) the validation period (1974-2003).  During the calibration,
the estimated $S_b$ based on CN is used, and $a$ is the only free parameter to be calibrated.  The
calibration is conducted by minimizing the absolute error of the observed and simulated mean
annual runoff through a global optimization method, i.e., Shuffled Complex Evolution Method
(Duan et al., 1992).  As shown in Figure 5b, most of the calibrated $a$ are smaller than the estimated
$a$ based on soil data only.  The performance of predicted mean annual runoff (during the validation
period) is improved with the calibrated shape parameter (Figure 5c).  The average of absolute error
for the mean annual runoff is 7.1%.

The overestimation of shape parameter based on the soil porosity data underestimates the

spatial variability of soil water storage capacity compared with the calibrated one as shown in
Figure 4a for the Fox River watershed and Figure 4b for the Spoon River watershed.  The slope at
the normalized storage capacity around 1 for the estimated shape parameter is higher than that for
the calibrated one.  Therefore, the calibrated shape parameter indicates a larger spatial variability.
The underestimation of the spatial heterogeneity of soil water storage capacity could be resulted





from neglecting the effect of land surface and bedrock topography which cannot be referred from
the soil database (gSSURGO) where the point-scale soil thickness is around 2 m.

To explore the impact of land surface topography on the spatial distribution of soil water

storage capacity, the soil data (i.e., porosity) is combined with the Height Above the Nearest
Drainage (HAND) method proposed by Gao et al. (2019).  HAND is the vertical elevation
difference from a point to its nearest drainage point.  The distribution of HAND was used for
estimating the shape parameter of the spatial distribution of storage capacity.  Therefore, the
HAND method uses land surface topography data only for estimating the shape parameter.  In our
analysis, the porosity of the soil beyond the bottom layer in the soil database is assigned with the
same value as the bottom layer.  For example, if the HAND for a grid cell is 10.0 m and the porosity
and depth of the bottom soil layer in the gSSURGO database is 0.2 and 2.0 m, respectively, the
porosity for the soil from 2.0 m to 10.0 m depth is assigned with 0.2.  Finally, the total volume of
pores is calculated for each grid cell based on the soil porosity obtained from the gSSURGO
database and the HAND value based on land surface topography.

Figure 6 shows the porosity-HAND based CDF of normalized soil water storage capacity

for the Maquoketa River in Iowa (gauge #05418500).  The stream initiation threshold used for
calculating HAND is 40 km$^2$ which is 1% of the maximum flow accumulation (Maidment, 2002).
The threshold affects the value of HAND but this is beyond the scope of this paper.  The best fit
value of $a$ for the porosity-HAND based CDF is 1.779, which overestimates the spatial variability
of storage capacity compared with the calibrated shape parameter ($a$=1.905).  This is due to the
assumption of the HAND method that the bedrock between a specific point and its nearest drainage
point is horizontal and intercepts with the channel bed.  However, the bedrock topography may
have various slopes in a watershed (Troch et al., 2002).  Therefore, the true value of $a$ (indicated





by the calibrated one) potentially falls between the $a$ obtained from soil data and the $a$ based on
soil and HAND. The bedrock topography from observation or models is needed to accurately
estimate the shape parameter.
**4. Conclusion**

A mean annual water balance model based on the concept of saturation excess runoff

generation is used for diagnosing the potential for nonparametric modeling of mean annual runoff
in ungauged basins. The model takes the effect of climate variability into account explicitly since
it is driven by daily precipitation and potential evapotranspiration at the daily time step. The
distribution function, which leads to the SCS curve number method, is used for describing the
spatial distribution of soil water storage capacity. The mean (i.e., average soil water storage
capacity) and the shape parameter (i.e., the spatial variability of soil storage capacity over the
watershed) of the distribution function can be estimated from the available data. Based on the
linkage of the distribution function and the SCS curve number method, a new method based on
the existing observed data of watershed characteristics is proposed for estimating the average soil
water storage capacity. The average soil water storage capacity ($S_b$), as one of the parameters in
the model, was estimated as a function of climate aridity index and curve number which is
calculated based on land cover and soil data.

The developed mean annual water balance was applied to diagnose the estimation of shape

parameter ($a$) in this study. The shape parameter, describing the spatial variation of soil water
storage capacity, was first estimated based on the porosity and soil thickness data in the soil
database (gSSURGO). The estimated values of $a$ were tested in 35 watersheds. The results
showed that the model with the estimated values of $S_b$ and $a$ underestimated the mean annual
runoff by 11.8% on average over all the study watersheds. The underestimation of runoff is mainly



caused by the underestimation of the spatial heterogeneity of soil thickness over the watershed.
The Height Above the Nearest Drainage (HAND) was then calculated as the total soil thickness
for estimating the total volume of the pore space. The result showed that topography is of great
importance for determining the spatial variability of soil water storage capacity. The estimated
shape parameter from porosity-HAND overestimated the spatial variability of the storage capacity
compared with the calibrated $a$, which may result from the assumed bedrock in the HAND method.
Future research will investigate alternative methods for better estimating the spatial variability of
soil water storage capacity over watersheds and test them in the proposed mean annual water
balance model.

**Data availability**
The soil and land use data used in this paper are provided in the references. Daily precipitation,
streamflow, and temperature data are downloaded from
ftp://hydrology.nws.noaa.gov/pub/gcip/mopex/US_Data/.

**Author contributions**
DW designed the analyses. YG and LY conducted the analyses. YG and DW wrote the paper.
LY and NC edited the paper.

**Competing interests**
The authors declare that they have no conflict of interest.

**Acknowledgements**




This research was funded in part by National Science Foundation (award CBET-1804770) and
Florida Department of Transportation.

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


Table 1: The USGS gage stations, climate aridity index, the estimated potential maximum
retention of curve number method ($S_{CN}$), and the average soil water storage capacity ($S_b$) for the
study watersheds.

| Index | Station Name | State | USGS Gauge Number | Climate Aridity Index | $S_{CN}$ (mm) | $S_b$ (mm) |
|---|---|---|---|---|---|---|
| 1 | Susquehanna River | NY | 01503000 | 0.69 | 100 | 862 |
| 2 | Chemung River | NY | 01531000 | 0.84 | 95 | 518 |
| 3 | Juniata River | PA | 01567000 | 0.85 | 134 | 714 |
| 4 | Rappahannock River | VA | 01668000 | 0.85 | 152 | 792 |
| 5 | Yadkin River | NC | 02116500 | 0.71 | 153 | 1221 |
| 6 | Chattahoochee River | GA | 02339500 | 0.69 | 182 | 1559 |
| 7 | Escambia River | FL | 02375500 | 0.73 | 143 | 1075 |
| 8 | Allegheny River | NY | 03011020 | 0.68 | 153 | 1369 |
| 9 | New River | VA | 03168000 | 0.69 | 177 | 1494 |
| 10 | Great Miami River | OH | 03274000 | 0.89 | 63 | 301 |
| 11 | Eel River | IN | 03328500 | 0.92 | 68 | 304 |
| 12 | East Fork White River | IN | 03364000 | 0.83 | 68 | 378 |
| 13 | Little Wabash River | IL | 03381500 | 0.96 | 68 | 279 |
| 14 | Fox River | WI | 04073500 | 1.12 | 162 | 520 |
| 15 | Auglaize River | OH | 04191500 | 0.98 | 56 | 225 |
| 16 | Maquoketa River | IA | 05418500 | 1.19 | 72 | 209 |
| 17 | Wapsipinicon River | IA | 05422000 | 1.16 | 69 | 210 |
| 18 | Rock River | WI | 05430500 | 1.11 | 98 | 316 |
| 19 | Pecatonica River | IL | 05435500 | 1.11 | 66 | 214 |
| 20 | Kishwaukee River | IL | 05440000 | 1.03 | 70 | 255 |
| 21 | Green River | IL | 05447500 | 1.10 | 75 | 247 |
| 22 | Iowa River | IA | 05454500 | 1.18 | 65 | 191 |
| 23 | Cedar River | IA | 05458500 | 1.17 | 65 | 193 |
| 24 | Kankakee River | IL | 05520500 | 0.93 | 101 | 448 |
| 25 | Fox River | IL | 05552500 | 1.04 | 88 | 321 |
| 26 | Spoon River | IL | 05570000 | 1.12 | 71 | 227 |
| 27 | Kaskaskia River | IL | 05592500 | 0.99 | 67 | 263 |
| 28 | Blue River | KS | 06884400 | 1.70 | 74 | 127 |
| 29 | Thompson River | MO | 06899500 | 1.16 | 65 | 195 |
| 30 | Meramec River | MO | 07019000 | 0.95 | 109 | 460 |
| 31 | Chikaskia River | OK | 07152000 | 1.82 | 77 | 121 |
| 32 | Neosho River | KS | 07183000 | 1.42 | 63 | 140 |
| 33 | Deep Fork River | OK | 07243500 | 1.40 | 87 | 197 |
| 34 | Neches River | TX | 08033500 | 1.14 | 174 | 540 |
| 35 | Elm Fork Trinity River | TX | 08055500 | 1.63 | 87 | 159 |






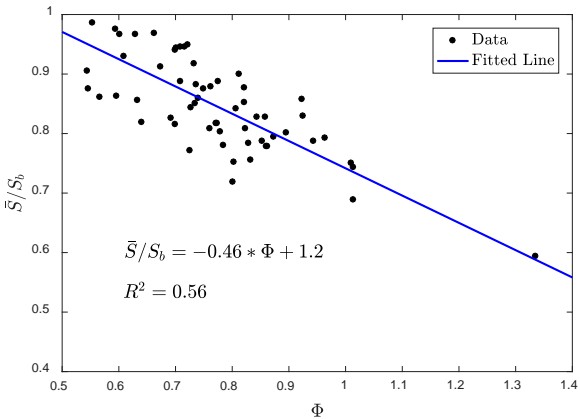


Figure 1: The degree of saturation $\left(\frac{\bar{S}}{S_b}\right)$ under long-term average climate versus climate aridity

index $(\Phi)$.




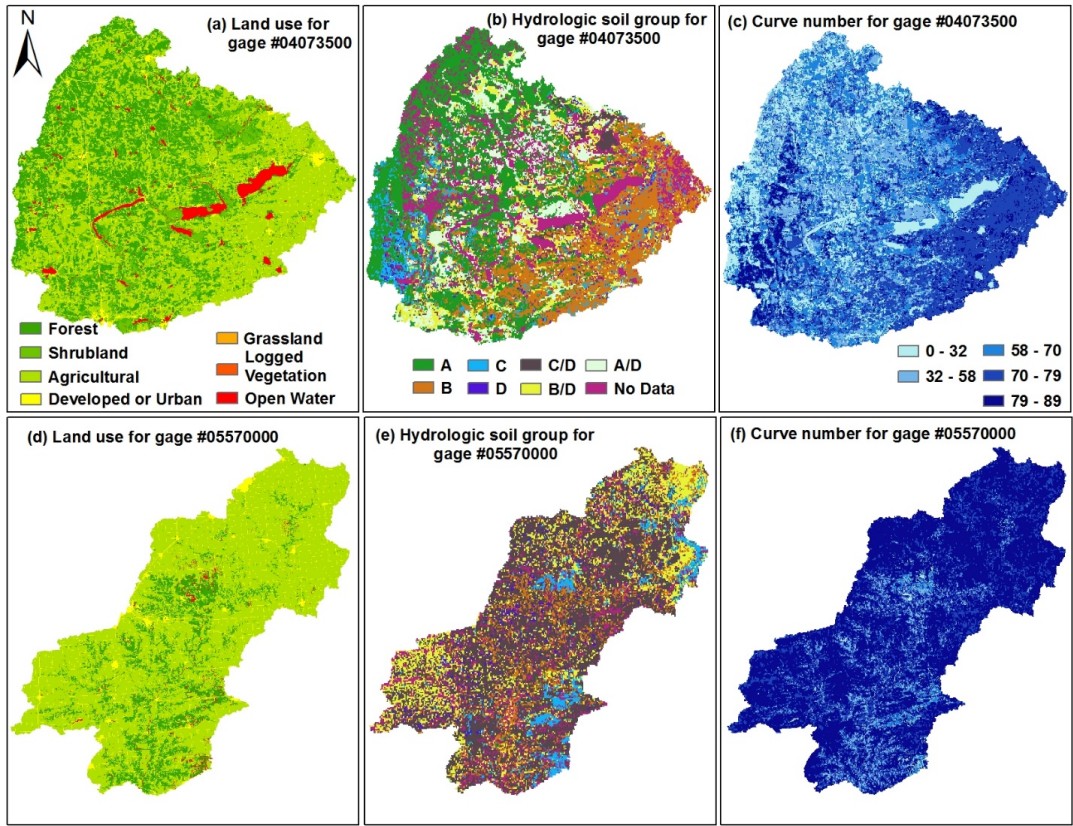

Figure 2: The spatial distribution of land use and land cover for Fox River watershed in
Wisconsin (a) and Spoon River watershed in Illinois (d), the hydrologic soil groups for Fox
River watershed (b) and Spoon River watershed (e), and the curve numbers for Fox River
watershed (c) and Spoon River watershed (f).





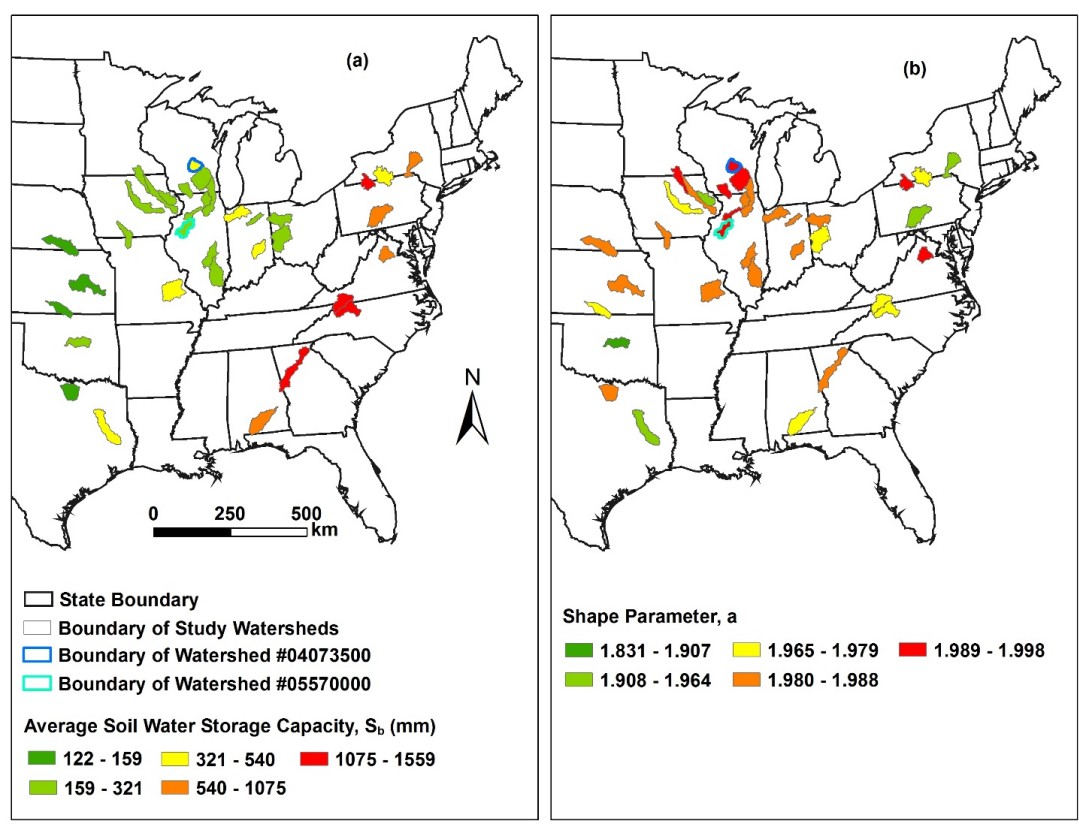

Figure 3: The estimated average soil water storage capacity ($S_b$) as a function of $S_{CN}$ and climate
aridity index (a) and shape parameter from soil data (b).






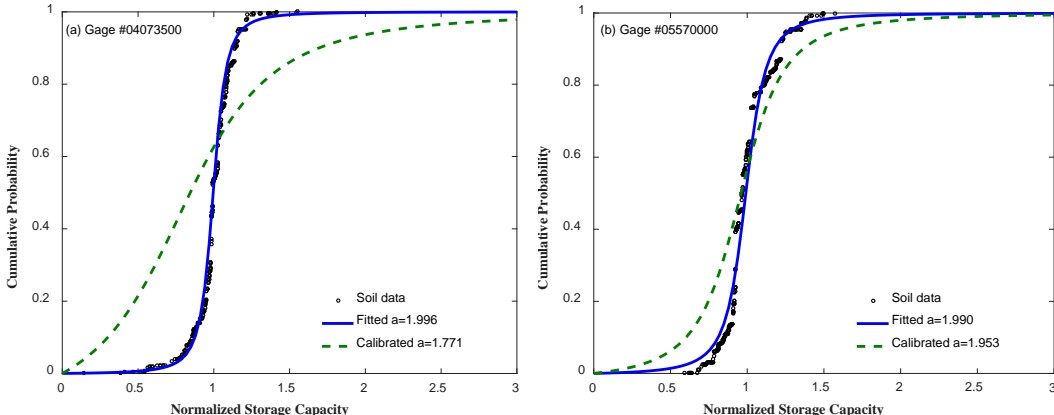


Figure 4: The estimated shape parameter for the spatial distribution of soil water storage capacity
based on soil data and the calibrated shape parameter based on mean annual water balance in the
Fox River watershed (a) and the Spoon River watershed (b).





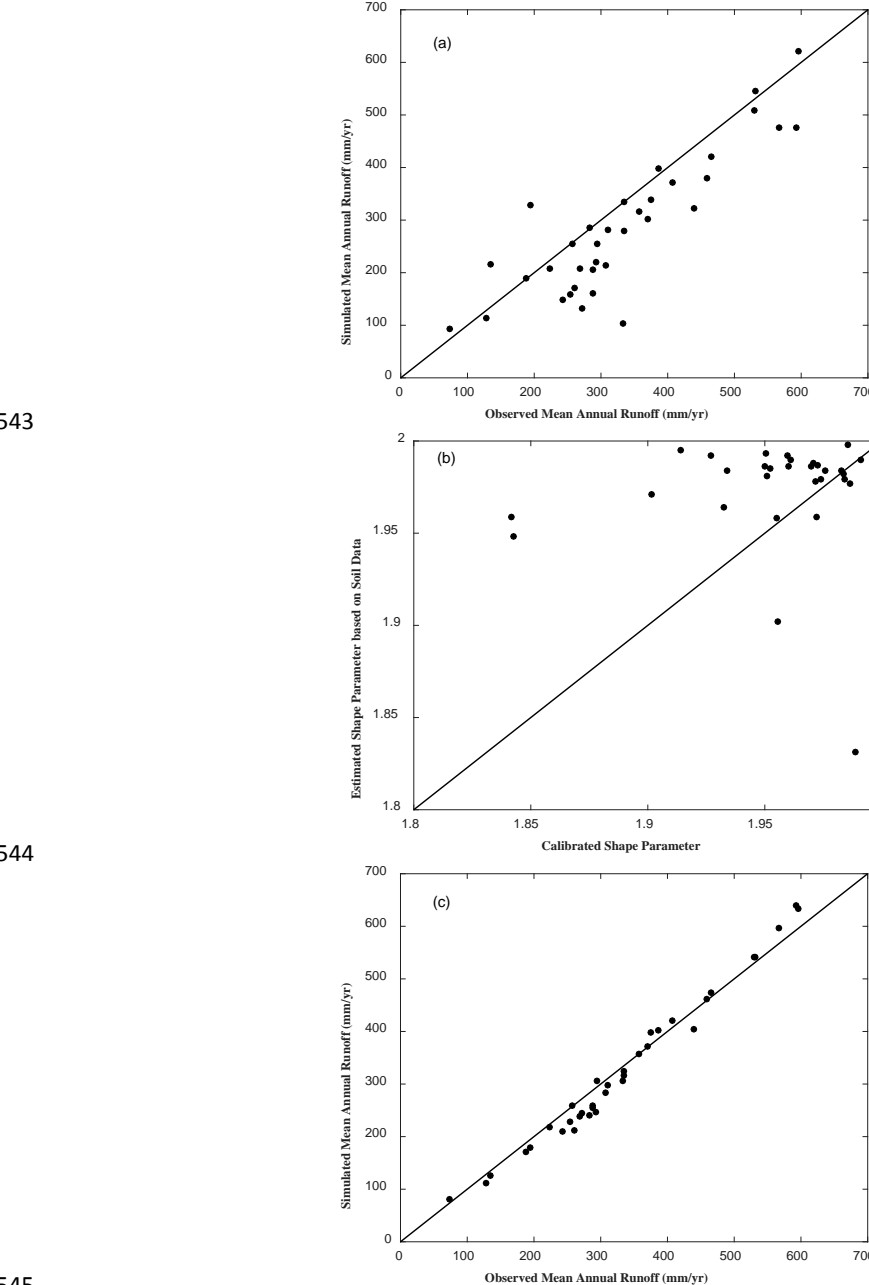

Figure 5: (a) Observed versus simulated mean annual runoff using shape parameter based on
soil data; (b) Soil data-based versus calibrated shape parameter; and (c) Observed versus
simulated mean annual runoff using shape parameter based on calibration.



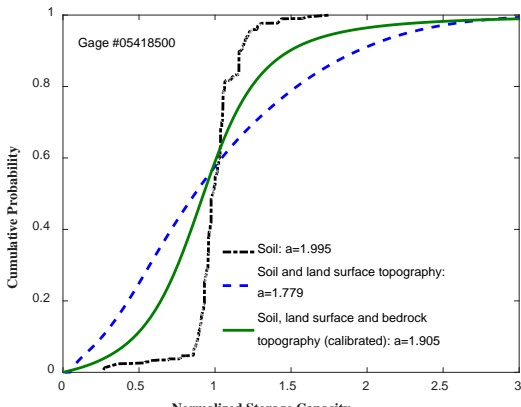


Figure 6: The effects of soil, land surface topography, and bedrock topography on the shape
parameter of the spatial distribution of soil water storage capacity.
