# Peer review of "Diagnosis toward predicting mean annual runoff in ungauged basins"

_Hydrology and Earth System Sciences, 2020_

## Referee Comment (RC1) · Anonymous Referee #1 · 10 Aug 2020

The authors aims to develop nonparametric mean annual water balance model for predicting mean annual runoff in ungauged basins, which often remains a challenge for hydrologists. The manuscript can advance prediction in ungaged basins, and it is relevant to HESS. I have following comments that needs to be discussed (addressed) at this stage.

(1) How the soil water storage is determined? It varies at seasonal scale. How does it will affect your analysis? It is worth to highlight following article that developed a three-parameter streamflow elasticity model as a function of precipitation, potential evap-oration, and change in groundwater storage applicable at both seasonal and annual scales. https://hess.copernicus.org/articles/20/2545/2016/

(2) What do mean by Climate variability in your study? does it mean distribution of

climate variables, for example, distribution of rainy days with in the season. This type of analysis are important and they have a direct influence on the soil water storage. This can be discussed as a scope of the future work. The magnitude and seasonality of the climate variables affects water availability (storage). This may be included as a future scope of the work. Please see this article: https://www.nature.com/articles/s41467-020-16757-w

(3) Are you using SCS method to find the infiltration loss? Does this loss is connected to shallow water storage?

(4) Baseflow plays an important role in the runoff analysis. Are you including this factor in your analysis. Can addition of the seasonal baseflow characteristics will improve the results?

(5) How the curve numbers are derived? Did you derive the composite curve numbers, i.e., one value for a watershed?

(6) How the bedrock topography are determined?

(7) I assume the shape parameter is kept constant for a given watershed, and it is calculated based by creating a time series based on the spatial (gridded) soil water capacity values. How the shape parameters are calculated? For example, Maximum Likelihood methods?? Do you think the parameter uncertainty (range) will affect the mean flow?

Line 98-100: Can be revised to make it simple.

---

## Referee Comment (RC2) · Anonymous Referee #2 · 19 Aug 2020

This manuscript tried to parameterize the two parameters of the mean annual balance equation by relating their values with the controlling factors, in order to develop a model to estimate mean annual runoff in ungauged basins. It is an interesting topic and suitable for HESS. However, I have several comments as follow. 1. It isn't clear which equation is the water balance model that was developed for estimating mean annual runoff. 2. As shown in Figure 5(b), there is a large difference and low correlation between the estimated shape parameter and the calibrated one. At the same time, Figure 5(a) shows that the model has a fair estimation of mean annual runoff with the estimated shape parameter. I guess that the model has a low sensitivity to the shape parameter. I suggest a sensitivity analysis on the parameter. Also, it is necessary to evaluate the improvement due to the parameterization from soil characteristics as

given in Section 2.2.2, since it is a relatively complicated process. In addition, I suggest that some statistical indicators should be given in Figure 5. 3. In Lines 142-148, the authors pointed out the effect of climate variability on water balance, but it isn't clear how to deal with the effect of climate variability in the developed model. In addition, previous studies reported that many factors, such as vegetation, catchment slope and etc., have an impact on water balance. I am not sure whether such factors have more lager impact on water balance than the spatial variability of storage capacity has. There is a possibility that their impacts can be attributed to the impact of the distribution of soil water storage capacity. More analysis and discussions are required.

---

## Referee Comment (RC3) · Anonymous Referee #3 · 3 Sep 2020

Wang and Gao et al conducted a study to develop a nonparametric mean annual water balance model for prediction in ungauged basins. They found that climate and topography play essential roles determining the storage capacity and its shape. I found this study is quite interesting and fits the scope of HESS. Relevant studies should be encouraged to understand and diagnose the impacts of different features on runoff generation in different time scales and their connections. Here I have several comments for the authors to consider for further improving the quality:

1. Why did the authors only use 35 catchments in this study? There are over 400 catchments in MOPEX data. Please clarify the reasons to exclude most catchments.

2. Line 73-74. I cannot follow this sentence. Please rephrase it.

[Figure]

3. Line 243. The Sb in Chattahoochee River watershed reaches to 1870mm. This value is too large, which let me doubt the physical meaning of the Sb parameter.

―――――――――――――――

---

## Referee Comment (RC4) · Anonymous Referee #1 · 8 Sep 2020

I appreciate the reviewers for answering my comments very diligently. I am recommending for acceptance.
* * *

---

## Author Comment (AC1) · 8 Sep 2020

Dear Reviewer: Thank you for your comments. Our responses to the comments are listed below: (1) How the soil water storage is determined? It varies at seasonal scale. How does it will affect your analysis? It is worth to highlight following article that developed a three parameter streamflow elasticity model as a function of precipitation, potential evaporation, and change in groundwater storage applicable at both seasonal and annual scales. https://hess.copernicus.org/articles/20/2545/2016/

Soil water storage capacity in this study is referred to as the maximum storage capacity from the land surface to the bedrock; therefore, it is considered as a static variable. The effective storage capacity or the remaining storage capacity could vary temporally due

to the dynamics of groundwater storage as shown in Konapala and Mishra (2016). The definition of the soil water storage capacity will be clarified in the revised manuscript.

Konapala, G., and Mishra, A. K. : Three-parameter-based streamflow elasticity model: Application to MOPEX basins in the USA at annual and seasonal scales., Hydrol. Earth Syst. Sci., 20, 2545-2556, https://doi.org/10.5194/hess-20-2545-2016.

(2) What do mean by Climate variability in your study? does it mean distribution of climate variables, for example, distribution of rainy days with in the season. This type of analysis are important and they have a direct influence on the soil water storage. This can be discussed as a scope of the future work. The magnitude and seasonality of the climate variables affects water availability (storage). This may be included as a future scope of the work. Please see this article: https://www.nature.com/articles/s41467-020-16757-w

Following Yao et al. (2020), the climate variability in this study is defined as the temporal variations of the precipitation (P) and potential evapotranspiration (Ep), including their intra-monthly, intra-annual, and inter-annual variations. For example, the deviations of daily P or Ep from its monthly mean values are defined as the intra-monthly variations. The definition of the climate variability will be included in the revised manuscript. In addition, we totally agree with you that the distribution of rainy days, the magnitude and the seasonality of climate variables have direct impacts on the soil water storage. In the revised manuscript, we will include the discussion on quantifying the climate variabilities, e.g., the distribution of rainy days, the mean and the seasonality of climate, and exploring the impacts of these characteristics on soil water storage capacity as a scope of our future work. Yao, L., Libera, D. A., Kheimi, M., Sankarasubramanian, A., and Wang, D (2020).: The roles of climate forcing and its variability on streamflow at daily, monthly, annual, and long‐term scales. Water Resour. Res., 55, e2020WR027111. https://doi.org/10.1029/2020WR027111.

(3) Are you using SCS method to find the infiltration loss? Does this loss is connected

to shallow water storage?

Yes. Infiltration loss is computed by Equation (2) which leads to the proportionality relationship of SCS method. The value of infiltration loss is dependent on the shallow water storage condition, which affects the remaining storage capacity. The "normal antecedent moisture" in the SCS curve number method is treated as the storage at the long-term steady-state condition. Therefore, the maximum storage capacity is the sum (Equation (5)) of storage capacity computed by the SCS curve number (Equation (4)) and long-term average storage.

(4) Baseflow plays an important role in the runoff analysis. Are you including this factor in your analysis. Can addition of the seasonal baseflow characteristics will improve the results?

We agree that baseflow plays an important role in total runoff which includes baseflow and surface runoff. But this research is focused on total runoff; therefore, baseflow is not explored separately in this study. One the other hand, the seasonal characteristics of baseflow are results of climate seasonality, which is implicitly included in the daily climate input. This will be clarified in the revised manuscript.

(5) How the curve numbers are derived? Did you derive the composite curve numbers, i.e., one value for a watershed?

Yes, each watershed has one curve number, which is the average curve number over the grid cells within the entire watershed. For each grid cell, the curve number is obtained based on land use and land cover and hydrologic soil group as introduced in Section 2.2.1. The composite curve number for each watershed will be clarified in the revised manuscript.

(6) How the bedrock topography are determined?

The bedrock topography data of the study catchments are not available from observations in this study; therefore, we used a hypothetical bedrock topography obtained

through Height Above the Nearest Drainage (HAND) method which assumes that the bedrock of each hillslope is horizontal and the bedrock elevation equals the elevation of the drainage point.

(7) I assume the shape parameter is kept constant for a given watershed, and it is calculated based by creating a time series based on the spatial (gridded) soil water capacity values. How the shape parameters are calculated? For example, Maximum Likelihood methods?? Do you think the parameter uncertainty (range) will affect the mean flow?

Yes, the shape parameter is kept constant for a given watershed. While, it is calculated by creating the spatial soil water capacity values at the long-term averaged antecedent soil moisture condition. A nonlinear programming solver using derivative-free method, i.e., Matlab function "fminsearch", was used to calculate the optimal shape parameter by minimizing the root mean square error (RMSE). The method will be clarified in the revised manuscript. Yes, the parameter uncertainty will affect the mean annual runoff which can be seen by comparing Figure 5a and 5c. For each catchment, the value of the average soil water storage capacity is same between these two figures, and the different simulation performance is only caused by the shape parameter. Clearly, the shape parameter could affect the mean annual runoff. In the revise manuscript, the sensitivity analysis of the mean annual runoff to the shape parameter will be conducted.

(8) Line 98-100: Can be revised to make it simple.

Thanks. This sentence would be revised as: The mean of the distribution is estimated from curve number and climate because the soil water storage capacity consists of the antecedent soil water storage and the potential maximum soil moisture retention which can be calculated through SCS curve number method.

Please also note the supplement to this comment:
https://hess.copernicus.org/preprints/hess-2020-353/hess-2020-353-AC1-

supplement.pdf

---

## Author Comment (AC2) · 8 Sep 2020

Reply to Referee #2: This manuscript tried to parameterize the two parameters of the mean annual balance equation by relating their values with the controlling factors, in order to develop a model to estimate mean annual runoff in ungauged basins. It is an interesting topic and suitable for HESS. However, I have several comments as follow.

Thank you very much for your comments and suggestions. Our replies are listed as follow:

(1) It isn't clear which equation is the water balance model that was developed for estimating mean annual runoff.

Thank you for pointing out the problem. The mean annual runoff is computed by the

difference of mean annual precipitation and mean annual evaporation which is computed by aggregating the daily evaporation calculated by Equation (3). This will be clarified and the equation for mean annual runoff will be presented explicitly in the revised manuscript.

(2) As shown in Figure 5(b), there is a large difference and low correlation between the estimated shape parameter and the calibrated one. At the same time, Figure 5(a) shows that the model has a fair estimation of mean annual runoff with the estimated shape parameter. I guess that the model has a low sensitivity to the shape parameter. I suggest a sensitivity analysis on the parameter. Also, it is necessary to evaluate the improvement due to the parameterization from soil characteristics as given in Section 2.2.2, since it is a relatively complicated process. In addition, I suggest that some statistical indicators should be given in Figure 5. 3.

Thank you for your suggestion. The narrow ranges of the axes may give us the impression that the difference between the estimated shape parameter and the calibrated one are large, while actually the mean difference is 0.06 which is small considered that the range of the shape parameter is from 0 to 2. The sensitivity analysis of the mean annual runoff to the shape parameter will be conducted in the revised manuscript, and statistical indicators will also be calculated for Figure 5. For the parameterization in Section 2.2.2, it is a new method proposed in this study to quantify the spatial heterogeneity of the soil water storage capacity, which is then discussed in Section 3 on how to improve the estimation by considering more details of the bedrock information, therefore, the focus of this study is not the improvement of the shape parameter parameterization from the soil characteristics.

(3) In Lines 142-148, the authors pointed out the effect of climate variability on water balance, but it isn't clear how to deal with the effect of climate variability in the developed model. In addition, previous studies reported that many factors, such as vegetation, catchment slope and etc., have an impact on water balance. I am not sure whether such factors have more lager impact on water balance than the spatial

variability of storage capacity has. There is a possibility that their impacts can be attributed to the impact of the distribution of soil water storage capacity. More analysis and discussions are required.

We are sorry for the confusion. Different from traditional mean annual water balance models which take the mean annual precipitation (P) and potential evapotranspiration (Ep) as climate inputs, our model is forced by the observed daily P and Ep; therefore, the effects of the climate variability, including the intra-monthly, intra-annual, and inter-annual climate variability are automatically included. In the revised manuscript, we will clarify how to deal with the effect of climate variability when we introduce the structure of the developed model in Section 2.1. For the other factors such as vegetation and catchment slope, we agree that their impacts can attribute to the distribution of soil water storage capacity as a result of catchment coevolution. The land surface topography, i.e., DEM, is one of the controlling factors for determining the soil thickness in this study; therefore, the topographic characteristics including the catchment slope has been considered through DEM data. To further explore the impact of catchment topographic features, we will add a discussion in the revised manuscript on determining the shape parameter of the soil storage capacity through the spatial variability of the topographic wetness index. For the impact of the vegetation on the soil water storage capacity distribution, it will be included as a future scope of our work.

Please also note the supplement to this comment:
https://hess.copernicus.org/preprints/hess-2020-353/hess-2020-353-AC2-supplement.pdf
* * *

---

## Author Comment (AC3) · 8 Sep 2020

Reply to Referee #3: Wang and Gao et al conducted a study to develop a nonparametric mean annual water balance model for prediction in ungauged basins. They found that climate and topography play essential roles determining the storage capacity and its shape. I found this study is quite interesting and fits the scope of HESS. Relevant studies should be encouraged to understand and diagnose the impacts of different features on runoff generation in different time scales and their connections. Here I have several comments for the authors to consider for further improving the quality:

We thank the reviewer for this positive feedback. Our responses to your comments are listed below.

[Figure]

(1) Why did the authors only use 35 catchments in this study? There are over 400 catchments in MOPEX data. Please clarify the reasons to exclude most catchments.

The 35 watersheds are selected considering the data availability including soil (hydrologic soil group), land cover and land use, DEM as well as the minimum snow effect and human activities. The data processing demand is also a consideration for selecting the limited number of watersheds. We think that the number of watersheds is sufficient for diagnosing the data requirement for estimating long-term runoff in ungagged basins, for example, the importance of bedrock data. The reasons will be clarified in the revised manuscript.

(2) Line 73-74. I cannot follow this sentence. Please rephrase it.

Thank you for pointing out the problem. This sentence would be revised as: It has also been suggested that the spatial variability of soil water storage capacity could suppress the actual evaporation because the maximum evaporation in areas with storage capacity less than Ep will smaller than Ep; therefore, the average evaporation over the entire catchment is smaller than Ep even though the average storage is greater than Ep, resulting in more runoff generation compared to the situation when the storage capacity is spatially uniform (Yao et al., 2020).

(3) Line 243. The Sb in Chattahoochee River watershed reaches to 1870mm. The value is to large, which let me doubt the physical meaning of the Sb parameter.

Sorry for the typo on the number of Sb in Chattahoochee River, and it should be 1559 mm. The physical meaning of Sb is the mean value of the soil water storage capacity over a catchment which is defined as the maximum storage from the land surface to the bedrock in this study rather than the storage capacity from shallow soils. Considering the maximum of soil water storage capacity could be 2000 mm from literature (Kollat et al., 2012), 1559 mm is considered to be reasonable in this study. To avoid this kind of the concern, the definition of the Sb will be clarified in the revised manuscript.

[Figure]

Kollat, J., Reed, P. M., and Wagener, T.: When are multiobjective calibration trade‐offs in hydrologic models meaningful?, Water Resour. Res., 48(3) https://doi.org/10.1029/2011WR011534.

Please also note the supplement to this comment:
https://hess.copernicus.org/preprints/hess-2020-353/hess-2020-353-AC3-supplement.pdf

---

## Author Response (AR1)

**Reply to Referee #1:**

Dear Reviewer: Thank you for your comments. Our responses to the comments are listed below:

(1) *How the soil water storage is determined? It varies at seasonal scale. How does it will affect your analysis? It is worth to highlight following article that developed a three parameter streamflow elasticity model as a function of precipitation, potential evaporation, and change in groundwater storage applicable at both seasonal and annual scales.*
*https://hess.copernicus.org/articles/20/2545/2016/*

Thank you for the comment. Soil water storage capacity in this study is referred to as the maximum storage capacity from the land surface to the bedrock; therefore, it is considered as a static variable. The effective storage capacity or the remaining storage capacity could vary temporally due to the dynamics of groundwater storage as shown in *Konapala and Mishra* (2016). The definition of the soil water storage capacity has been clarified on Lines 60-61 in the revised manuscript:

Lines 60-61: "Soil water storage capacity is the maximum storage capacity from land surface to bedrock, which exerts a powerful control on mean annual runoff (Konapala and Mishra, 2016)."

"Konapala, G., and Mishra, A. K.: Three-parameter-based streamflow elasticity model: Application to MOPEX basins in the USA at annual and seasonal scales., Hydrol. Earth Syst. Sci., 20, 2545-2556, https://doi.org/10.5194/hess-20-2545-2016."

(2) *What do mean by Climate variability in your study? does it mean distribution of climate variables, for example, distribution of rainy days within the season. This type of analysis are important and they have a direct influence on the soil water storage. This can be discussed as a scope of the future work. The magnitude and seasonality of the climate variables affects water availability (storage). This may be included as a future scope of the work. Please see this article: https://www.nature.com/articles/s41467-020-16757-w*

Thank you for the comment. Following *Yao et al*. (2020), the climate variability in this study is defined as the temporal variations of precipitation ($P$) and potential evapotranspiration ($Ep$), including their intra-monthly, intra-annual, and inter-annual variations. For example, the deviations of daily $P$ or $Ep$ from its monthly mean values are defined as the intra-monthly variations. The definition of climate variability has been included in the revised manuscript on Lines 113-118. In addition, we totally agree with you that the distribution of rainy days, the magnitude and the seasonality of climate variables have direct impacts on soil water storage, and we have included them as a scope of our future work on Lines 392-395 in the revised manuscript:

Lines 113-118: "Climate variability is defined as the temporal variations of precipitation ($P$) and potential evapotranspiration ($E_p$), including their intra-monthly, intra-annual, and inter-annual variations. For example, the deviations of daily $P$ or $E_p$ from its monthly mean values are defined as the intra-monthly variations (Yao et al., 2020). As discussed in the Introduction section, the mean annual runoff model takes daily precipitation and potential evaporation as inputs, therefore, climate variability is explicitly included in the model."

"Yao, L., Libera, D. A., Kheimi, M., Sankarasubramanian, A., and Wang, D (2020): The roles of climate forcing and its variability on streamflow at daily, monthly, annual, and long-term scales. Water Resour. Res., 55, e2020WR027111. https://doi.org/10.1029/2020WR027111."

Lines 392-395: "Future research will investigate alternative methods for better estimating the spatial variability of soil water storage capacity over watersheds, and quantify the impacts of vegetation and climate variability (e.g., distribution of rainy days, the magnitude and the seasonality of climate variables)."

(3) *Are you using SCS method to find the infiltration loss? Does this loss is connected to shallow water storage?*

Yes. Infiltration loss is computed by Equation (2) which leads to the proportionality relationship of SCS method. The value of infiltration loss is dependent on the shallow water storage condition, which affects the remaining storage capacity. The "normal antecedent moisture" in the SCS curve number method is treated as the storage at the long-term steady-state condition. Therefore, the maximum storage capacity is the sum (Equation (7)) of storage capacity computed by the SCS curve number (Equation (6)) and long-term average storage.

(4) *Baseflow plays an important role in the runoff analysis. Are you including this factor in your analysis. Can addition of the seasonal baseflow characteristics will improve the results?*

We agree that baseflow plays an important role in total runoff which includes baseflow and surface runoff. However, this research is focused on total runoff; therefore, baseflow is not explored separately in this study. On the other hand, the seasonal characteristics of baseflow are results of climate seasonality, which is implicitly included in the daily climate input. This has been clarified on Lines 153-155 in the revised manuscript:

Lines 153-155: "Note that the mean annual runoff includes surface runoff and baseflow, and both are impacted by climate variability (e.g., intra-annual variability) (Berghuijs et al., 2014; Fan et al., 2007)."

"Fan, Y., Miguez-Macho, G., Weaver, C. P., Walko, R., and Robock, A: Incorporating water table dynamics in climate modeling: 1. Water table observations and equilibrium water table simulations, J. Geophys. Res., 112, D10125, doi:10.1029/2006JD008111, 2007.

Berghuijs, W. R., Sivapalan, M., Woods, R. A., and Savenije, H. H.: Patterns of similarity of seasonal water balances: A window into streamflow variability over a range of time scales, Water Resour. Res., 50(7), 5638-5661, https://doi.org/10.1002/2014WR015692, 2014."

(5) *How the curve numbers are derived? Did you derive the composite curve numbers, i.e., one value for a watershed?*

Yes, each watershed has one curve number, which is the average curve number over the grid cells within the entire watershed. For each grid cell, the curve number is obtained based on land use and land cover and hydrologic soil group as introduced in Section 2.2.1. The composite curve number for each watershed has been clarified in the revised manuscript:

Lines 175-176: "where CN is the composite curve number based on land use and land cover (LULC) and hydrologic soil group (HSG) for each watershed."

(6) *How the bedrock topography are determined?*

The bedrock topography data of the study catchments are not available from observations in this study; therefore, we used a hypothetical bedrock topography obtained through Height Above the Nearest Drainage (HAND) method which assumes that the bedrock of each hillslope is horizontal and the bedrock elevation equals the elevation of the drainage point. This has been clarified on Lines 354-355 in the revised manuscript:

Lines 354-355: "This is due to the assumption of the HAND method that the bedrock between a specific point and its nearest drainage point is horizontal and intercepts with the channel bed."

(7) *I assume the shape parameter is kept constant for a given watershed, and it is calculated based by creating a time series based on the spatial (gridded) soil water capacity values. How the shape parameters are calculated? For example, Maximum Likelihood methods?? Do you think the parameter uncertainty (range) will affect the mean flow?*

Yes, the shape parameter is kept constant for a given watershed. While, it is calculated by creating the spatial soil water capacity values under the long-term averaged antecedent soil moisture condition. A nonlinear programming solver using derivative-free method, i.e., Matlab function "fminsearch", was used to calculate the optimal shape parameter by minimizing the root mean square error (RMSE). The method has been clarified on Lines 227-230 in the revised manuscript. For the parameter uncertainty, its impact on the mean annual runoff can be seen by comparing Figures 5a and 5c. The value of the average soil water storage capacity of each catchment is same between these two figures, and the different simulation performance is only caused by the shape parameter. Clearly, the shape parameter could largely affect the mean annual runoff. In the revise manuscript, the sensitivity of mean annual runoff to the shape parameter has been conducted, and is shown in the new figure, i.e., Figure 2, and the clarification has been added on Lines 230-238 in the revised manuscript:

Lines 227-230: "The shape parameter $a$ is then estimated by fitting the point-scale storage capacity data obtained from Equation (11). A nonlinear programming solver using derivative-free method (i.e., Matlab function "fminsearch") was used to calculate the optimal shape parameter by minimizing the root mean square error (RMSE)."

Lines 230-238: "To demonstrate the sensitivity of mean annual runoff to the value of shape parameter, Figure 2 presents mean annual runoff versus shape parameter based on the mean annual water balance (Yao et al., 2020). It can be found that mean annual runoff decreases significantly as shape parameter increases, especially when shape parameter approaches its upper limit (i.e., 2). The negative relationship between mean annual runoff and shape parameter can be attributed to the fact that the larger shape parameter indicates that less watershed area has small values of point-scale storage capacity (Wang, 2018) and more precipitation could be retained underground for evaporation."

[Figure]

Figure 2: The sensitivity of mean annual runoff ($Q$) to the value of shape parameter ($a$).

(8) *Line 98-100: Can be revised to make it simple.*

Thanks. This sentence has been revised on Lines 104-107 in the revised manuscript:

"The mean soil water storage capacity is estimated from curve number and climate because soil water storage capacity consists of the antecedent soil water storage and the potential maximum soil moisture retention which can be calculated through SCS curve number method."

**Reply to Referee #2:**

*This manuscript tried to parameterize the two parameters of the mean annual balance equation by relating their values with the controlling factors, in order to develop a model to estimate mean annual runoff in ungauged basins. It is an interesting topic and suitable for HESS. However, I have several comments as follow.*

Thank you very much for your comments and suggestions. Our replies are listed as follow:

(1) *It isn't clear which equation is the water balance model that was developed for estimating mean annual runoff.*

Thank you for pointing out the problem. The mean annual runoff is computed by the difference of mean annual precipitation and mean annual evaporation which is computed by aggregating the daily evaporation calculated by Equation (3). This has been clarified and the equation for mean annual runoff has been presented explicitly on Lines 147-153 in the revised manuscript:

Lines 147-153: "Mean annual evaporation ($\bar{E}$) is computed by aggregating the daily evaporation, and mean annual runoff ($\bar{Q}$) is computed as the difference of mean annual precipitation and evaporation:

$$\bar{E} = \frac{\sum_{y=1}^{Y}\sum_{d=1}^{D_y} E_d}{Y} \qquad (4)$$

$$\bar{Q} = P - \bar{E} \qquad (5)$$

where, $Y$ is the number of years, and $D_y$ is the number of days in y^th year; $y$ and $d$ represent the y^th year and $d^{th}$ day, respectively."

(2) *As shown in Figure 5(b), there is a large difference and low correlation between the estimated shape parameter and the calibrated one. At the same time, Figure 5(a) shows that the model has a fair estimation of mean annual runoff with the estimated shape parameter. I guess that the model has a low sensitivity to the shape parameter. I suggest a sensitivity analysis on the parameter. Also, it is necessary to evaluate the improvement due to the parameterization from soil characteristics as given in Section 2.2.2, since it is a relatively complicated process. In addition, I suggest that some statistical indicators should be given in Figure 5. 3.*

Thank you for your suggestion. The narrow ranges of the axes may give us the impression that the difference between the estimated shape parameter and the calibrated one are large, while actually the mean difference is 0.06 which is small considered that the range of the shape parameter is from 0 to 2. The sensitivity analysis of the mean annual runoff to the shape parameter has been conducted and shown in the new figure (i.e., Figure 2), and the clarification has been added on Lines 230-238 in the revised manuscript. The coefficients of determination ($R^2$) have been calculated for Figure 6 (Figure 5 in the original version) in the revised manuscript. For the parameterization in Section 2.2.2, it is a new method proposed in this study to quantify the spatial heterogeneity of the soil water storage capacity, which is then discussed in Section 3 on how to improve the estimation by considering more details of the bedrock information, therefore, the focus of this study is not the improvement of the shape parameter parameterization from the soil characteristics.

Lines 230-238: "To demonstrate the sensitivity of mean annual runoff to the value of shape parameter, Figure 2 presents mean annual runoff versus shape parameter based on the mean annual water balance (Yao et al., 2020).  It can be found that mean annual runoff decreases significantly as shape parameter increases, especially when shape parameter approaches its upper limit (i.e., 2). The negative relationship between mean annual runoff and shape parameter can be attributed to the fact that the larger shape parameter indicates that less watershed area has small values of point-scale storage capacity (Wang, 2018) and more precipitation could be retained underground for evaporation."

[Figure]

[Figure]

Figure 6: (a) Observed versus simulated mean annual runoff using shape parameter based on soil data; (b) Soil data-based versus calibrated shape parameter; and (c) Observed versus simulated mean annual runoff using shape parameter based on calibration.

*(3) In Lines 142-148, the authors pointed out the effect of climate variability on water balance, but it isn't clear how to deal with the effect of climate variability in the developed model. In addition, previous studies reported that many factors, such as vegetation, catchment slope and etc., have an impact on water balance. I am not sure whether such factors have more lager impact on water balance than the spatial variability of storage capacity has. There is a possibility that their impacts can be attributed to the impact of the distribution of soil water storage capacity. More analysis and discussions are required.*

We are sorry for the confusion. Different from traditional mean annual water balance models which take the mean annual precipitation ($P$) and potential evapotranspiration ($E_p$) as climate inputs, our model is forced by the observed daily $P$ and $E_p$; therefore, the effects of the climate variability, including the intra-monthly, intra-annual, and inter-annual climate variability are explicitly included. In the revised manuscript, we have clarified how to deal with the effect of climate variability when we introduce the structure of the developed model in Section 2.1 (Lines 113-118). For the other factors such as vegetation and catchment slope, we agree that their impacts can attribute to the distribution of soil water storage capacity as a result of catchment coevolution. The land surface topography (i.e., DEM) is one of the controlling factors for determining the soil thickness in this study; therefore, the topographic characteristics including the catchment slope has been considered through DEM data. To further explore the impact of catchment topographic features, we have added a discussion on determining the shape parameter of the soil storage capacity through the spatial variability of the topographic wetness index in Lines 340-347 and 359-364 in the revised manuscript. For the impact of vegetation on the soil water storage capacity distribution, it has been included as a future scope of our work on Lines 392-395 in the revised manuscript:

Lines 113-118: "Climate variability is defined as the temporal variations of precipitation ($P$) and potential evapotranspiration ($E_p$), including their intra-monthly, intra-annual, and inter-annual variations. For example, the deviations of daily $P$ or $E_p$ from its monthly mean values are defined as the intra-monthly variations (Yao et al., 2020). As discussed in the Introduction section, the mean annual runoff model takes daily precipitation and potential evaporation as inputs, therefore, climate variability is explicitly included in the model."

Lines 340-347: "The control of land surface topography on the hydrologic process has also been widely quantified through topographic wetness index (TWI) of TOPMODEL (Beven and Kirkby, 1979). The spatial variability of soil storage capacity based on the TOPMODEL assumption has been demonstrated as a beneficial representation of the conceptual model (Sivapalan et al., 1997). Therefore, the heterogeneity of TWI in a watershed was proposed to be another surrogate of the heterogeneity of the soil storage capacity in this study, and the shape parameter estimated by fitting TWI against Equation (12) through minimizing the root mean square error (RMSE) for the Maquoketa River in Iowa was compared with those obtained from other methods."

Lines 359-364: "The dashed dot red line in Figure 7 displays the CDF of the normalized soil storage capacity based on TWI, and the corresponding value of $a$ is 1.967. The TWI-based $a$ value also presents a larger spatial variability than that derived from soil data solely, confirming the importance of topography in determining the heterogeneity of soil water storage capacity. The deviation of the TWI-based $a$ value from its calibrated counterpart could be due to the fact that the bedrock topography is not considered in TWI."

[Figure]

Figure 7: The effects of soil, land surface topography, bedrock topography, and topographic wetness index (TWI) on the shape parameter of the spatial distribution of soil water storage capacity.

Lines 392-395: "Future research will investigate alternative methods for better estimating the spatial variability of soil water storage capacity over watersheds, and quantify the impacts of vegetation and climate variability (e.g., distribution of rainy days, the magnitude and the seasonality of climate variables)."

**Reply to Referee #3:**

*Wang and Gao et al conducted a study to develop a nonparametric mean annual water balance model for prediction in ungauged basins. They found that climate and topography play essential roles determining the storage capacity and its shape. I found this study is quite interesting and fits the scope of HESS. Relevant studies should be encouraged to understand and diagnose the impacts of different features on runoff generation in different time scales and their connections. Here I have several comments for the authors to consider for further improving the quality:*

We thank the reviewer for this positive feedback. Our responses to your comments are listed below.

(1) *Why did the authors only use 35 catchments in this study? There are over 400 catchments in MOPEX data. Please clarify the reasons to exclude most catchments.*

The 35 watersheds are selected considering the data availability including soil (hydrologic soil group), land cover and land use, DEM as well as the minimum snow effect and human activities. The data processing demand is also a consideration for selecting the limited number of watersheds. We think that the number of watersheds is sufficient for diagnosing the data requirement for estimating long-term runoff in ungagged basins, for example, the importance of bedrock data. The reasons have been clarified in the revised manuscript on Lines 240-244:

"The number of 35 was determined due to the consideration of the data availability including soil (hydrologic soil group), land cover and land use, DEM as well as the minimum snow effect and human activities (Wang and Hejazi, 2011), and to keep the efforts of gSSURGO data processing to a reasonable level while still to have a sufficient number of sample of catchments."

   (2) *Line 73-74. I cannot follow this sentence. Please rephrase it.*

Thank you for pointing out the problem. This sentence has been revised on Line 74-80 in the revised manuscript:

"It has also been suggested that the spatial variability of soil water storage capacity could suppress the actual evaporation because the maximum evaporation in areas with soil water storage capacity less than $E_p$ will be smaller than $E_p$; therefore, the average evaporation over the entire catchment is smaller than $E_p$ even though the average storage is greater than $E_p$, resulting in more runoff generation compared to the situation when the soil water storage capacity is spatially uniform (Yao et al., 2020)."

(3) *Line 243. The Sb in Chattahoochee River watershed reaches to 1870mm. The value is to large, which let me doubt the physical meaning of the Sb parameter.*

Sorry for the typo on the number of $S_b$ in Chattahoochee River, and it should be 1559 mm. The value has been corrected on Lines 275-277 in the revised manuscript. The physical meaning of $S_b$ is the mean value of the soil water storage capacity over a catchment which is defined as the maximum storage from land surface to bedrock in this study rather than the storage capacity from shallow soils. Considering the maximum of soil water storage capacity could be 2000 mm from literature (*Kollat et al.*, 2012), 1559 mm is considered to be reasonable in this study. The definition of the $S_b$ has been clarified in the revised manuscript on Lines 187-189.

Lines 275-277: "As shown in Table 1, the estimated $S_b$ varies from 177 mm (Chikaskia River watershed) to 1559 mm (Chattahoochee River watershed) over the study watersheds."

Lines 187-189: "The physical meaning of $S_b$ is the mean value of the soil water storage capacity over a watershed which is defined as the maximum storage from land surface to bedrock in this study rather than the storage capacity from shallow soils."

Kollat, J., Reed, P. M., and Wagener, T.: When are multiobjective calibration trade-offs in hydrologic models meaningful?, Water Resour. Res., 48(3) https://doi.org/10.1029/2011WR011534.

[revised manuscript text omitted]